# Reinforcement Learning for Control with Multiple Frequencies

**Jongmin Lee[1], Byung-Jun Lee[1], Kee-Eung Kim[1,2]**
[1] School of Computing, KAIST, Republic of Korea
[2] Graduate School of AI, KAIST, Republic of Korea
{jmlee,bjlee}@ai.kaist.ac.kr, kekim@kaist.ac.kr

## Abstract

Many real-world sequential decision problems involve multiple action variables whose control frequencies are different, such that actions take their effects at different periods. While these problems can be formulated with the notion of multiple action persistences in factored-action MDP (FA-MDP), it is non-trivial to solve them efficiently since an action-persistent policy constructed from a stationary policy can be arbitrarily suboptimal, rendering solution methods for the standard FA-MDPs hardly applicable. In this paper, we formalize the problem of multiple control frequencies in RL and provide its efficient solution method. Our proposed method, Action-Persistent Policy Iteration (AP-PI), provides a theoretical guarantee on the convergence to an optimal solution while incurring only a factor of $|\mathcal{A}|$ increase in time complexity during policy improvement step, compared to the standard policy iteration for FA-MDPs. Extending this result, we present Action-Persistent Actor-Critic (AP-AC), a scalable RL algorithm for high-dimensional control tasks. In the experiments, we demonstrate that AP-AC significantly outperforms the baselines on several continuous control tasks and a traffic control simulation, which highlights the effectiveness of our method that directly optimizes the periodic non-stationary policy for tasks with multiple control frequencies.

## 1 Introduction

In recent years, reinforcement learning (RL) [23] has shown great promise in various domains, such as complex games [14, 21, 22] and high-dimensional continuous control [11, 19]. These problems have been mostly formulated as *discrete-time* Markov decision processes (MDPs) [17], assuming all decision variables are simultaneously determined at every time step. However, many real-world sequential decision-making problems involve multiple decision variables whose control frequencies are different by requirement. For example, when managing a financial portfolio of various assets, the frequency of rebalancing may need to be different for each asset, e.g. weekly for stock and monthly for real estate. Similarly, robotic systems typically consist of a number of controllers operating at different frequencies due to their system specification.

Different control frequencies can be formulated with the notion of different *action persistence* in the discrete-time factored-action MDP (FA-MDP), where the base time interval is determined by the reciprocal of the least common multiple of the control frequencies. However, while algorithms for *single* action persistence has been proposed in order to improve the empirical performance of online [9] or offline [13] RL agents, to the best of our knowledge, addressing *multiple* action persistences in RL has been mostly unexplored due to its difficulty involved in the non-stationarity nature of the optimal policy.

In this paper, we formalize the problem of multiple action persistences in FA-MDPs. We first show that any persistent policy induced by a stationary policy can be arbitrarily bad via a simple example.

Then, we introduce efficient methods for FA-MDPs that directly optimize a periodic non-stationary policy while circumventing the exponential growth of time complexity with respect to the periodicity of action persistence. We first present a tabular planning algorithm, Action-Persistent Policy Iteration (AP-PI), which provides the theoretical guarantee on the convergence to an optimal solution while incurring only a factor of $|\mathcal{A}|$ time complexity increase in the policy improvement step compared to the policy iteration for standard FA-MDPs. We then present Action-Persistent Actor-Critic (AP-AC), a scalable learning algorithm for high-dimensional tasks via practical approximations to AP-PI, with a neural network architecture designed to facilitate the direct optimization of a periodic non-stationary policy. In the experiments, we demonstrate that AP-AC significantly outperforms a number of baselines based on SAC, from the results on modified Mujoco continuous control benchmarks [3, 26] and the SUMO traffic control simulation [8], which highlights the effectiveness of our method that directly optimizes the periodic non-stationary policy for tasks with multiple control frequencies.

## 2 Preliminaries

We assume the environment modeled as discrete-time factored-action MDP (FA-MDP) $\mathcal{M} = \langle \mathcal{S}, \mathcal{A}, P, R, \gamma \rangle$ where $\mathcal{S}$ is the set of states $s$, $\mathcal{A}$ is the set of vector-represented actions $a = (a^1, \ldots, a^m)$, $P(s'|s, a) = \Pr(s_{t+1} = s'|s_t = s, a_t = a)$ is the transition probability, $R(s, a) \in \mathbb{R}$ is the immediate reward for taking action $a$ in state $s$, and $\gamma \in [0, 1)$ is the discount factor. A policy $\pi = (\pi_t)_{t \geq 0} \in \Pi$ is a sequence of functions where $\pi_t : \mathcal{H} \to \Delta(\mathcal{A})$ is a mapping from history $h_t = (s_0, a_0, \ldots, s_{t-1}, a_{t-1}, s_t)$ to a probability distribution over $\mathcal{A}$, $\pi_t(a_t|h_t) = \Pr(a_t|h_t)$. We call $\pi_t$ Markovian if $\pi_t$ depends only on the last state $s_t$ and call it stationary if $\pi_t$ does not depend on $t$. The policy $\pi_t$ is called deterministic if it maps from history to some action with probability 1 and can be denoted as $\pi_t : \mathcal{H} \to \mathcal{A}$. For simplicity, we will only consider the fully factorized policy $\pi_t(a_t^1, \ldots, a_t^m|h_t) = \prod_{k=1}^{m} \pi_t^k(a_t^k|h_t)$, which comprises the set $\Pi$ of all fully factorized policies. The action-value function $Q_t^\pi$ of policy $\pi$ is defined as $Q_t^\pi(s, a) = \mathbb{E}_\pi \left[ \sum_{\tau=t}^{\infty} \gamma^{\tau-t} R(s_\tau, a_\tau)|s_t = s, a_t = a \right]$.

We consider the sequential decision problem where each action variable $a^k$ has its own control frequency. The notion of control frequency can be formulated in terms of *action persistence* with FA-MDP $\mathcal{M}$ by considering how frequently $a^k$ should be decided in $\mathcal{M}$. Specifically, we let $c^k$ be the action persistence of $k$-th action variable $a^k$, i.e. $a^k$ is decided every $c^k$ time step in $\mathcal{M}$. The overall action persistence of the decision problem is then described as a vector $c = (c^1, \ldots, c^m) \in \mathbb{N}^m$. Finally, we define the $c$-persistent policy $\pi$ as follows:

**Definition 1.** ($c$-persistent policy) *Let $\pi = (\pi_t)_{t \geq 0} \in \Pi$ be a policy. Given the action persistence vector $c \in \mathbb{N}^m$, the $c$-persistent policy $\bar{\pi}_c = (\bar{\pi}_{c,t})_{t \geq 0}$ induced by $\pi$ is a non-stationary policy where*

$$\forall t, \ \bar{\pi}_{c,t}(a|h_t) = \prod_{k=1}^{m} \bar{\pi}_{c,t}^k(a^k|h_t) \ s.t. \ \bar{\pi}_{c,t}^k(a^k|h_t) = \begin{cases} \pi_t^k(a^k|h_t) & \text{if } t \bmod c^k = 0 \\ \delta_{a_{t-(t \bmod c^k)}^k}(a^k) & \text{otherwise} \end{cases} \tag{1}$$

*where $\delta_x(y) = 1$ if $x = y$ and $0$ otherwise. Additionally, we define the set of $c$-persistent policies $\Pi_c = \{(\bar{\pi}_{c,t})_{t \geq 0} : \pi \in \Pi\}$.*

Our goal is to find the $c$-persistent policy $\pi_c^*$ that maximizes expected cumulative rewards:

$$\bar{\pi}_c^* = \arg\max_{\bar{\pi} \in \Pi_c} \mathbb{E}_{\bar{\pi}} \left[ \sum_{t=0}^{\infty} \gamma^t R(s_t, a_t) \right] \tag{2}$$

**Remark.** When $c = (1, \ldots, 1)$, we have $\Pi_c = \Pi$. Thus, Eq. (2) is reduced to the standard objective function of FA-MDP, which is known to always have a deterministic and Markovian stationary policy as an optimal solution [17]. Also, the $c$-persistent policy of Definition 1 is different from the $k$-persistent policy [13] in that our definition considers *multiple* action persistences and is *not* limited by Markovian policy $\pi$ while [13] considers *single* action persistence and a non-stationary policy induced only by a *Markovian* policy.

The agent with $c$-persistent policy $\bar{\pi}_c$ induced by $\pi$ interacts with the environment as follows: At time step $t = 0$, all action variables are selected according to $\bar{\pi}_{c,0} = \pi_0$, i.e. $(a_0^1, \ldots, a_0^m) \sim \prod_{k=1}^{m} \pi_0^k(\cdot|h_0)$. Then, each action variable $a^k$ is kept *persistent* for the subsequent $c^k - 1$ time steps. At time step $t = c^k$, the action variable $a^k$ is set by $\bar{\pi}_{c,t}^k(\cdot|h_t) = \pi_t^k(\cdot|h_t)$, and continue into the next

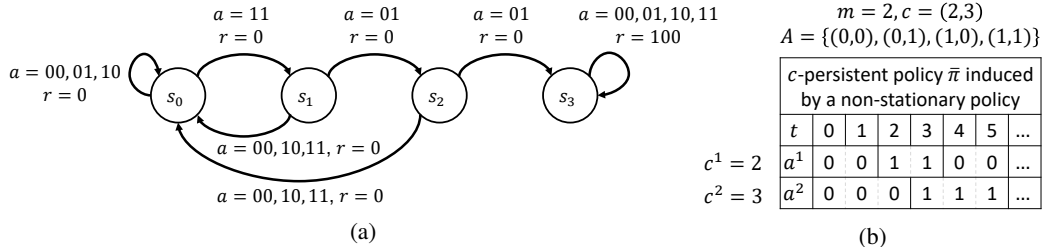

Figure 1: An illustrative example of FA-MDP with two action persistence where the initial state is $s_0$ and $\gamma = 0.95$. The arrows indicate transitions and rewards for each action. In this simple example, every $c$-persistent policy induced by any stationary policy is suboptimal.

time step. In other words, the agent decides the value for $a^k$ only at the time steps $t$ that are multiples of $c^k$, i.e. $t \bmod c^k = 0$. Figure 1b illustrates an example of $c$-persistent policy $\bar{\pi}_c$. For the remainder of this paper, we will omit the subscript $c$ in $\bar{\pi}_c$ for notational brevity if there is no confusion. All the proofs of theorems are available in the Appendix.

## 3 Action-Persistence in FA-MDPs

Finding the optimal policy via Eq. (2) is non-trivial since any $c$-persistent policy naively constructed from a *stationary* policy can be suboptimal, unlike in standard FA-MDPs where there always exists a stationary optimal policy. To see this, consider the FA-MDP depicted in Figure 1, where there are two action variables with action persistences 2 and 3, respectively. In this example task, in order to obtain a positive reward, the agent should take an action $a = (1, 1)$ at state $s_0$ to go to the rightmost state. However, when we use this to form a stationary deterministic policy with $\pi(s_0) = (1, 1)$ and construct a $c$-persistent policy in a naive manner, we see that the policy can never reach $s_3$ due to the inherent action persistence $c = (2, 3)$: The action $(1, 1)$ taken at $s_0$ when $t = 0$ will persist at the next time step $t = 1$ in $s_1$, making the agent go back to $s_0$. Then, the agent will select an action $(1, 1)$ again by $\pi(s_0)$, and this will be repeated forever. As a consequence, the agent visits only $s_0$ and $s_1$, and thus cannot reach the rightmost state. In contrast, the *non-stationary* deterministic policy $\bar{\pi}$ described in Figure 1b reaches $s_3$. Careful readers may notice that a $c$-persistent policy "projected" from some stationary but stochastic policy can eventually reach $s_3$, but its expected return is clearly less than the non-stationary deterministic policy in Figure 1b, thus suboptimal.

Therefore, obtaining a $c$-persistent policy by ignoring the action persistence requirement and solving the corresponding standard FA-MDP would not work. However, one can observe that the action persistence scheme is repeated periodically at every $L \triangleq \mathrm{LCM}(c^1, \dots, c^m)$ time steps. From this observation, a naive approach to solving Eq. (2) would be redefining the action space to have $L$-step actions as elements. After redefining the transition and reward function corresponding to these actions, standard solution methods for FA-MDP such as dynamic programming can be applied. Still, this approach not only has exponential time complexity with respect to $L$ due to the increase in the size of action space, i.e. $|\mathcal{A}|^L$, but also can be suboptimal unless the underlying transition dynamics is nearly deterministic due to the *open-loop* decision-making nature of $L$-step actions [27]. A more principled approach is to consider an $L$-Markovian policy that memorizes which action was taken during the last $L$ steps, but its straightforward conversion to the standard MDP via state augmentation still suffers from the exponential time complexity with respect to $L$.

### 3.1 Policy evaluation for $c$-persistent policy: $c$-persistent Bellman operators

As discussed in the previous section, augmenting state or action space for storing $L$-step information results in exponential complexity with respect to $L$. Instead, we take a more direct approach that optimizes the $c$-persistent policy via composition of Bellman operators within the space of $L$-periodic, non-stationary and deterministic policies $\Pi_L$:

$$\Pi_L = \{\pi \in \Pi : \forall t, \pi_t = \pi_{t+L} \text{ and } \pi_t : \mathcal{A} \times \mathcal{S} \to \mathcal{A}\} \tag{3}$$

We will later prove that there always exists an optimal policy for Eq. (2), which is induced by $\pi \in \Pi_L$. The policy in $\Pi_L$ will be denoted as $\pi = (\pi_0, \dots, \pi_{L-1})$ in the remainder of the paper.

As the first step of the derivation of our algorithm, we define function $\Gamma_{t,a}^c(a')$:

$$\Gamma_{t,a}^c(a') = (\bar{a}^1, \ldots \bar{a}^m) \text{ where } \bar{a}^k \triangleq \begin{cases} a^k & \text{if } t \bmod c^k \neq 0 \\ a'^k & \text{if } t \bmod c^k = 0 \end{cases} \tag{4}$$

which projects action $a'$ into a feasible action at time step $t$ if the action taken at $t-1$ is assumed to be $a$. This is done by extracting dimensions of "effectable" action variables at time step $t$ from $a'$ and extracting dimensions of "uneffectable" variables at time step $t$ from $a$.

For the $L$-periodic non-stationary deterministic policy $\pi = (\pi_0, \ldots, \pi_{L-1}) \in \Pi_L$, we first define the one-step $c$-persistent Bellman operator $\bar{\mathcal{T}}_t^\pi$ induced by $\pi$. Specifically, for $t \in \{0, \ldots, L-1\}$,

$$(\bar{\mathcal{T}}_t^\pi Q)(s,a) \triangleq R(s,a) + \gamma \mathbb{E}_{\substack{s' \sim P(s'|s,a) \\ a' = \pi_{t+1}(a,s')}} \left[ Q(s', \Gamma_{t+1,a}^c(a')) \right] \tag{5}$$

Then, we define an $L$-step $c$-persistent Bellman operator $\bar{H}_t^\pi$ by making the composition of $L$ one-step $c$-persistent Bellman operators:

$$(\bar{H}_0^\pi Q)(s,a) \triangleq (\bar{\mathcal{T}}_0^\pi \bar{\mathcal{T}}_1^\pi \cdots \bar{\mathcal{T}}_{L-2}^\pi \bar{\mathcal{T}}_{L-1}^\pi Q)(s,a) \tag{6}$$

$$(\bar{H}_1^\pi Q)(s,a) \triangleq (\bar{\mathcal{T}}_1^\pi \bar{\mathcal{T}}_2^\pi \cdots \bar{\mathcal{T}}_{L-1}^\pi \bar{\mathcal{T}}_0^\pi Q)(s,a)$$

$$\vdots$$

$$(\bar{H}_{L-1}^\pi Q)(s,a) \triangleq (\bar{\mathcal{T}}_{L-1}^\pi \bar{\mathcal{T}}_0^\pi \cdots \bar{\mathcal{T}}_{L-3}^\pi \bar{\mathcal{T}}_{L-2}^\pi Q)(s,a)$$

The following theorem and corollary state that each $L$-step $c$-persistent Bellman operator $\bar{H}_t^\pi$ is a contraction mapping, and each of the fixed points $Q_0^{\bar{\pi}}, \ldots, Q_{L-1}^{\bar{\pi}}$ has a recursive relationship with another by one-step $c$-persistent Bellman operators $\bar{\mathcal{T}}_0^\pi, \ldots, \bar{\mathcal{T}}_{L-1}^\pi$.

**Theorem 1.** *For all $t \in \{0, \ldots, L-1\}$, the $L$-step $c$-persistent Bellman operators $\bar{H}_t^\pi$ is $\gamma^L$-contraction with respect to infinity norm, thus $\bar{H}_t^\pi Q_t^{\bar{\pi}} = Q_t^{\bar{\pi}}$ has the unique fixed point solution. In other words, for any $Q_t^0 : \mathcal{S} \times \mathcal{A} \to \mathbb{R}$, define $Q_t^{n+1} = \bar{H}_t^\pi Q_t^n$. Then, the sequence $Q_t^n$ converges to $t$-th $c$-persistent value function of $\bar{\pi}$ as $n \to \infty$.*

**Corollary 1.** *$Q_t^{\bar{\pi}} = \bar{\mathcal{T}}_t^\pi Q_{(t+1) \bmod L}^{\bar{\pi}}$ holds for all $t \in \{0, \ldots, L-1\}$, thus $c$-persistent value functions can be obtained by repeatedly applying 1-step $c$-persistent backup in a $L$-cyclic manner.*

Note that the $c$-persistent value function of the policy $\pi$ obtained by $\bar{H}_t^\pi$, has the following form:

$$Q_t^{\bar{\pi}}(s,a) = \mathbb{E}_{\substack{\forall \tau, \, s_{\tau+1} \sim P(\cdot|s_\tau,a_\tau) \\ \bar{a}_{\tau+1} = \Gamma_{\tau+1,\bar{a}_\tau}^c(\pi_{\tau+1}(\bar{a}_\tau,s_{\tau+1}))}} \left[ \sum_{\tau=t}^\infty \gamma^{\tau-t} R(s_\tau, \bar{a}_\tau) \,\Big|\, s_t = s, \bar{a}_t = a \right] \tag{7}$$

which is obtained by unfolding the $L$-step $c$-persistent Bellman recursion from $\bar{H}_t^\pi$. Here, one can easily show that every action taken at every time step $t$, which is projected by $\Gamma_{t,\bar{a}}^c(\cdot)$, abides by $c$-persistence, by mathematical induction. As a result, $Q_t^{\bar{\pi}}(s,a)$ has the intended interpretable meaning, i.e. the expected sum of rewards that can be obtained when following the $c$-persistent policy $\bar{\pi}$ which is induced by $\pi$, except for the initial action $a$, starting from the state $s$ at time step $t$.

**Remark.** The time complexity of applying the one-step $c$-persistent Bellman backup $\bar{\mathcal{T}}_t^\pi$ of Eq. (5) for a deterministic policy $\pi$ is $O(|\mathcal{S}|^2|\mathcal{A}|)$ for each $t$, which is identical to the time complexity of the non-persistent standard Bellman backup.

Now, we have a complete policy evaluation operator for $c$-persistent policy induced by $L$-periodic non-stationary deterministic policy $\pi$.

### 3.2 Policy improvement for $c$-persistent policy

The remaining step for full policy iteration is policy improvement using $Q_t^{\bar{\pi}}(s,a)$.

**Theorem 2.** *Given a $L$-periodic, non-stationary, and deterministic policy $\pi = (\pi_0, \ldots, \pi_{L-1}) \in \Pi_L$, let $Q_t^{\bar{\pi}}$ be the $c$-persistent value of $\bar{\pi}$ denoted in Eq. (7). If we update the new policy $\pi^{\text{new}} = (\pi_0^{\text{new}}, \ldots, \pi_{L-1}^{\text{new}}) \in \Pi_L$ by*

$$\forall t, a, s', \, \pi_t^{\text{new}}(a, s') = \arg\max_{a'} Q_t^{\bar{\pi}}(s', \Gamma_{t,a}^c(a')) \tag{8}$$

*then $Q_t^{\bar{\pi}^{\text{new}}}(s,a) \geq Q_t^{\bar{\pi}}(s,a)$ holds for all $t, s, a$.*

**Remark.** The time complexity of policy improvement step defined by Eq. (8) is $O(|\mathcal{S}||\mathcal{A}|^2)$ for each $t$, which has $|\mathcal{A}|$ times worse time complexity compared to the standard non-persistent policy improvement whose complexity is $O(|\mathcal{S}||\mathcal{A}|)$. Note also that the new policy $\pi^{\text{new}}$ is not necessarily $c$-persistent, i.e. $\pi^{\text{new}} \notin \Pi_c$ is possible, but the performance of its inducing $c$-persistent policy is always improved.

Finally, Theorems 1 and 2 lead us to a full algorithm, action-persistent policy iteration (AP-PI). AP-PI iterates between $c$-persistent policy evaluation by Eq. (6) and the $c$-persistent policy improvement of Eq. (8), and it is guaranteed to converge to the optimal $c$-persistent policy $\bar{\pi}^* \in \Pi_c$. The pseudo-code of AP-PI can be found in Appendix D.

**Theorem 3.** *Starting from any $\bar{\pi}^0 \in \Pi_c$ induced by $L$-periodic non-stationary deterministic policy $\pi^0 \in \Pi_L$, the sequence of value functions $Q^{\bar{\pi}^n}$ and the improved policies $\bar{\pi}^{n+1}$ induced by $\pi^{n+1}$ converge to the optimal value function and the optimal $c$-persistent policy $\bar{\pi}^*$, i.e. $Q_t^{\bar{\pi}^*}(s,a) = \lim_{n\to\infty} Q_{t \bmod L}^{\bar{\pi}^n}(s,a) \geq Q_t^{\bar{\pi}}(s,a)$ for any $\bar{\pi} \in \Pi_c$, $t \in \mathbb{N}_0$, $s \in \mathcal{S}$, and $a \in \mathcal{A}$.*

**Corollary 2.** *There always exists a $c$-persistent optimal policy $\bar{\pi}_c^*$, which is induced by a $L$-periodic, non-stationary, and deterministic policy $\pi \in \Pi_L$.*

The policy $\bar{\pi}^* = (\bar{\pi}_0^*, \dots, \bar{\pi}_{L-1}^*)$ obtained by AP-PI is executed as follows. First, $\bar{a}$ is initialized randomly. Then, at every step $t$, $a_t = \Gamma_{t,\bar{a}}^c(\bar{\pi}_{t \bmod L}^*(\bar{a}, s_t))$ is executed, and $\bar{a}$ is updated by $\bar{a} \leftarrow a_t$.

To the best of our knowledge, AP-PI is the first algorithm that addresses *multiple* action persistences, extending the *single* action persistence model that has been recently analyzed in [13]. AP-PI can be readily made scalable using the actor-critic architecture, to cope with large action spaces such as continuous actions, which we describe in the next section. This is a non-trivial extension of Persistent Fitted Q-iteration (PFQI) [13] which only applies to finite action spaces with single action persistence.

# 4 Action-Persistent Actor-Critic

In this section, we present Action-Persistent Actor-Critic (AP-AC), an off-policy RL algorithm that can be applied to high-dimensional tasks via practical approximation to AP-PI. AP-AC extends Soft Actor-Critic (SAC) [4] to perform iterative optimization of the parametric models of an $L$-periodic non-stationary policy (i.e. actor), and its $c$-persistent action-value function (i.e. critic). We assume that the action persistence vector $c = (c_1, \dots, c_m)$ is given as a part of the environment specification.

As discussed in Section 3, the optimal $c$-persistent policy $\bar{\pi}$ can be induced by an $L$-periodic non-stationary policy $\pi = (\pi_0, \dots, \pi_{L-1})$, where $\pi_t : \mathcal{A} \times \mathcal{S} \to \Delta(\mathcal{A})$ for all $t$. The corresponding optimal value function is also represented by the $L$-periodic action-value function $Q^{\bar{\pi}} = (Q_0^{\bar{\pi}}, \dots, Q_{L-1}^{\bar{\pi}})$ with $Q_t^{\bar{\pi}} : \mathcal{S} \times \mathcal{A} \to \mathbb{R}$ for all $t$. We exploit this structure of the optimal solution in the neural network architecture. Specifically, the parameterized actor network $\pi_\phi(\bar{a}, s)$ and the critic network $Q_\theta(s, a)$ are designed to have $L$ heads, whose $t$-th head represents $\pi_t$ and $Q_t^{\bar{\pi}}$ respectively, thus sharing the parameters of the lower layers among different $t$. The $t$-th head of the critic recursively references the $((t + 1) \bmod L)$-th head for the target value, reflecting the result of Corollary 1.

The $c$-persistent value function is trained to minimize the squared temporal difference error:

$$J_Q(\theta) = \frac{1}{L} \sum_{t=0}^{L-1} \mathbb{E}_{\substack{(s,a,r,s')\sim\mathcal{D} \\ a'\sim\pi_{\phi,(t+1)\bmod L}(\cdot|a,s')}} \left[ \left( Q_{\theta,t}(s,a) - y_t(a,r,s',a') \right)^2 \right] \quad (9)$$

$$\text{s.t. } y_t(a,r,s',a') = r + \gamma Q_{\bar{\theta},(t+1)\bmod L}\left(s', \Gamma_{t+1,a}^c(a')\right) - \alpha \log \pi_{\phi,(t+1)\bmod L}(a'|a,s'),$$

where $\mathcal{D}$ denotes the replay buffer, $\bar{\theta}$ is the parameters of the target network, and $\Gamma_{t,a}^c(a')$ is the action projection function defined in Eq. (4). This objective function is obtained from Eq. (5) with an (optional) entropy regularization term $\alpha \log \pi_{\phi,t}(a'|s')$, following the SAC formulation. Note that every term in Eq. (9) is agnostic to the actual time step when $(s,a,r,s')$ was collected, which is due to the way we calculate $y_t$ using $Q_{(t+1)\bmod L}$ and $\Gamma$. Thus every $(s,a,r,s')$ sample in $\mathcal{D}$ can be used to train $Q_{\theta,t}$ regardless of $t$.

The policy parameters are then optimized by maximizing:

$$J_\pi(\phi) = \frac{1}{L} \sum_{t=0}^{L-1} \mathbb{E}_{\substack{(s,a,r,s')\sim\mathcal{D} \\ a'\sim\pi_{\phi,t}(\cdot|a,s')}} \left[ Q_{\theta,t}(s', \Gamma_{t,a}^c(a')) - \alpha \log \pi_{\phi,t}(a'|a,s') \right] \quad (10)$$

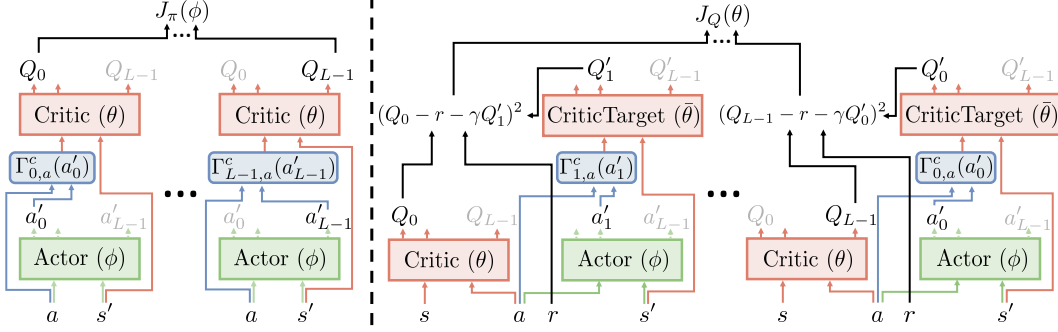

Figure 2: Overview of the network architectures and computational graphs for AP-AC training, given the $(s, a, r, s')$ sample. The left figure corresponds to actor training of, Eq. (10), and the right figure to critic training of Eq. (9).

where the (optional) $\alpha \log \pi_{\phi,t}(a'|a, s')$ term comes from SAC formulation. In essence, maximizing $J_\pi(\phi)$ with respect to $\phi$ corresponds to $c$-persistent policy improvement by implementing Eq. (8) approximately. As with the case with critic, every term in Eq. (10) is agnostic to the actual time step $t$ of when $(s, a, r, s')$ was collected, thus every sample in $\mathcal{D}$ can be used to train $\pi_{\phi,t}$ for all $t$. The overall network architecture and the computational graph for training AP-AC are visualized in Figure 2. In order to obtain lower-variance gradient estimate $\hat{\nabla}_\phi J_\pi(\phi)$, we adopt the exact reparameterization [7] for continuous action tasks and the relaxed reparameterization with Gumbel-softmax [5, 12] for discrete action tasks. The rest of the design choices follows that of SAC such as the clipped double Q trick and soft target update. The pseudo-code for AP-AC can be found in Appendix E.

## 5 Related Works

**Action Repetition in RL**    Recent deep RL algorithms have adopted action repetition to improve learning efficiency by reducing control granularity. Static action repetition, which repeats the same action over a fixed $k$ time step, has been widely adopted in both on-policy [15] and off-policy [14] RL. Dynamic action repetition [9, 20] has also been explored to further improve the learning efficiency of online RL agents by adaptively changing the time scale of repeating actions per state. Recently, the notion of a single action-persistence has been formalized by introducing persistent Bellman operators, and its corresponding offline RL algorithm has been proposed along with a heuristic method for finding good persistence for empirical performance [13]. In contrast to the existing works that consider a *single* action-persistence, we deal with arbitrarily *multiple* action-persistence where each decision variable has its own persistence, and our goal is to provide an efficient solution method for the given action persistence $c$ rather than finding a proper $c$ to speed up learning.

**Temporal Abstraction in RL**    The notion of action persistence is also naturally related to temporally abstract actions [16, 25] and semi-MDP framework [2]. Specifically, persisting actions with multiple frequencies can be seen as a particular instance of a semi-Markov *option* as follows: initiation set is the set of all states $\mathcal{I} = \mathcal{S}$, an internal policy is $c$-persistent $\pi \in \Pi_c$, and the termination condition is defined as $\beta(h_t) = \mathbb{1}_{\{t \bmod L = 0\}}$. Then, our off-policy learning scheme that exploits every transition sample to update every timestep's actor and critic in Eq. (9-10) can also be understood as an intra-option learning [24] method in the constructed semi-Markov option framework. Still, the cardinality of the set of possible options has an exponential growth with respect to $L$, thus obtaining an optimal policy over the set of options will be computationally inefficient compared to AP-PI that enjoys a linear complexity with respect to $L$.

## 6 Experiments

We conducted a set of experiments in order to evaluate the effectiveness of AP-AC on high-dimensional tasks with different control frequencies. To the best of our knowledge, this work is the first to address multiple control frequencies in RL. Since there are no existing RL methods

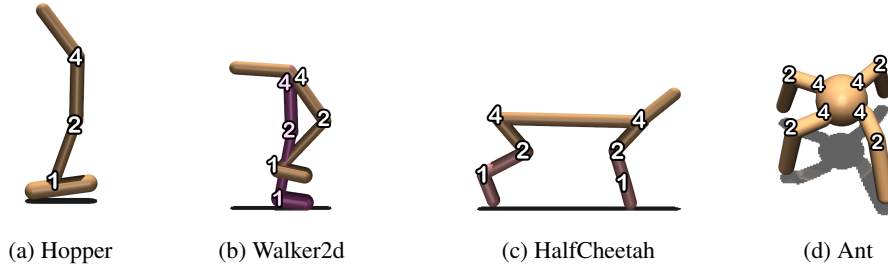

| (a) Hopper | (b) Walker2d | (c) HalfCheetah | (d) Ant |

Figure 3: Description of Mujoco continuous control tasks with multiple control frequencies. The number on each joint indicates the degree of persistence of each action. In other words, the action persistence vector for each domain is given as: $c = (4, 2, 1)$ for Hopper, $c = (4, 2, 1, 4, 2, 1)$ for Walker2d and HalfCheetah, and $c = (4, 2, 4, 2, 4, 2, 4, 2)$ for Ant.

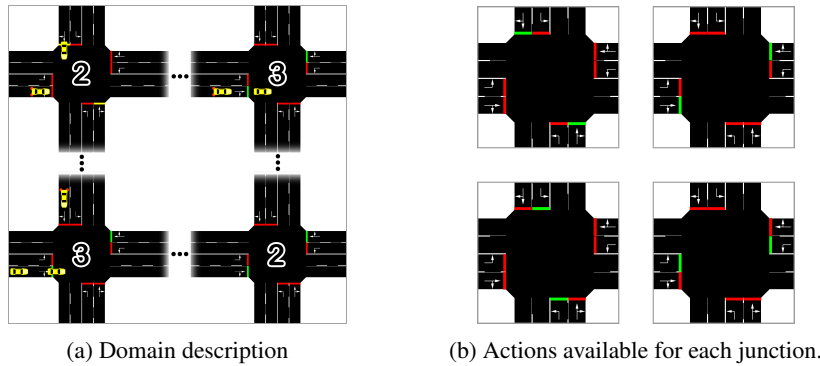

| (a) Domain description | (b) Actions available for each junction. |

Figure 4: Description of traffic light control problem. (a) We consider a 2x2 traffic simulation consisting of two rows and two columns of roads, where the traffic lights for the top-left and the bottom-right junctions have action persistence 2 while other junctions have action persistence 3, i.e. $c = (2, 3, 3, 2)$. (b) For each junction, there are $2^2$ actions (i.e. the traffic light) where the green light allows the traffic to proceed and the red light prohibits any traffic from proceeding.

designed for multiple control frequencies, we take the variants of SAC as baselines for performance comparison, which are listed as follows: (1) **SAC**: this agent is trained on the standard *non-persistent* environment, while being evaluated on the environment where the action-persistence is enforced. This is intended to show the suboptimality of simply projecting a stationary policy to an action-persistent policy. (2) **SAC in AP-Env**: this agent is trained and evaluated on the action-persistent version of the environment, using the standard RL algorithm. This is to demonstrate the suboptimality of a stationary Markovian policy. (3) **SAC-L**: this agent takes a current observation, past $L$ actions, and the one-hot indicator of the current time step ($t \bmod L$), which are sufficient for the optimal decision-making for the corresponding state augmentation approach discussed in Section 3. Still, this does not exploit the structure of the $c$-persistent optimal solution such as periodically recurrent policy/value representation and can take redundant information which is not fully compact. As a consequence, it is expected to show relatively weak performance. (4) **SAC-L-compact**: this agent takes a current observation, the last action which was *actually taken*, and the one-hot indicator of the current time step ($t \bmod L$), which is a compact representation of SAC-L. Still, this is unable to exploit every transition sample to update every timestep's actor and critic, while AP-AC is capable of doing it in Eq. (9-10). Therefore, it is expected to be less sample-efficient than AP-AC.

We conducted experiments on both continuous and discrete tasks, which will be detailed in the following section. The experimental setups including hyperparameters can be found in Appendix G.

## 6.1 Task description

**Mujoco tasks (continuous action space)** In many real-world situations, complex robotic systems consist of a number of controllers whose operating control frequencies vary due to the system

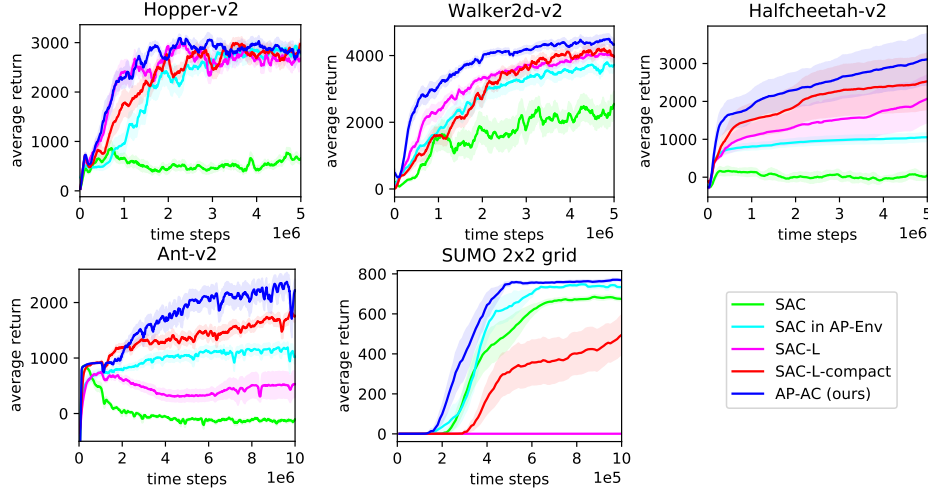

Figure 5: The results of experiments on the continuous control benchmarks and the traffic light control problems, which are averaged over 10 trials. The shaded area represents the standard error.

specification. In order to simulate this setting, we first conduct experiments on four OpenAI Gym continuous control tasks based on the Mujoco physics simulator [3, 26], where the controllable joints are modified to have different action persistence. Figure 3 depicts the detailed experimental setup for different action persistence for each task. For Hopper and Walker2d, action persistence for the thigh(s), the leg(s), and the foot(feet) are set to 4, 2, and 1 respectively. For HalfCheetah, action persistence for the thighs, the shins, and the feet are set to 4, 2, and 1 respectively. Finally, for Ant, the persistence for the hips and ankles are set to 4 and 2. We represent the policy $\pi_t$ as the Gaussian with diagonal covariance matrix and $\tanh$-squashing function to bound the output in range $[-1, 1]$ for each dimension [4].

**Traffic light control (discrete action space)**  We also tested AP-AC on a traffic control task, a realistic discrete-action sequential decision scenario with action persistence: in the traffic system, the control frequency of each traffic light can be different, for example depending on the number lanes and the speed limit. We use SUMO (Simulation of Urban MObility) [8] as the traffic simulator and SUMO-RL [1] for the environment interface. The specific instance we use is the implementation of 2X2GRID in SUMO-RL, which is depicted in Figure 4. The goal is to manipulate traffic lights located at each junction to improve the overall traffic flow, where the vehicles are generated randomly with a probability of 0.1 for every second at the end of the road.

The observation for each junction consists of the following four types of values: (1) the current traffic light status, represented by (4D one-hot), (2) the elapsed time from the current traffic light status, normalized within $[0, 1]$ (1D), (3) the density of all vehicles for each lane (8D), and (4) the density of stopped vehicles for each lane (8D). Therefore, the overall dimension of the observation space is $4 \times 21 = 81$. The action space is described in Figure 4b. The reward in the range $[0, 1]$ is defined to be $\min_{i \in \{1,2,3,4\}} 1/(\text{waiting time of junction } i)$, with the goal of improving traffic flow of the junction of heaviest traffic. The length of episodes is 1000. We use a factorized (relaxed) categorical distribution to represent the policy with discrete action space, i.e. $\pi_{\phi,t}(a|\cdot) = \prod_{k=1}^{4} \text{Cat}(a^k|p_\phi^k(\cdot))$ where $p_\phi(\cdot)$ denotes the probability vector with size 4. Though the cardinality of the entire joint action space is $|A| = 4^4 = 256$, the input and output dimensions to represent actions in the actor/critic networks are $4 \times 4 = 16$ (i.e. four one-hot vectors with size 4) since we are assuming fully factorized policies.

## 6.2 Results

We performed deterministic evaluation for each algorithm every 10K time steps, i.e. the peformance of the mean policy for continuous control and the greedy policy with respect to the categorical probabilities for the traffic control. The results are presented in Figure 5.

Since **SAC** (colored in green) is optimized for the non-persistent environment, its naive projection to the $c$-persistent policy suffers from severe performance degradation. In contrast, **SAC in AP-Env** (colored in cyan) interacts with the $c$-persistent environment directly while optimizing a stationary Markovian policy. Still, as discussed in Section 3, stationary Markovian policies can be suboptimal in general, which resulted in performing worse than AP-AC. **SAC-L** (colored in magenta) takes the past $L$-step actions and indicator of the current time step ($t \mod L$), which is sufficient information for optimal $c$-persistent decision-making. Nonetheless, it does not exploit the structure of optimal $c$-persistent solution and can take redundant information since not all the past $L$-step actions are required for optimal decision-making, resulting in inefficient learning. This can be observed from the results that as the action dimension increases (Hopper (3) $\rightarrow$ Walker/Halfcheetah (6) $\rightarrow$ Ant (8) $\rightarrow$ SUMO2x2GRID (16)), the performance of SAC-L gets relatively worse. **SAC-L-compact** (colored in red) takes the last action actually taken, and indicator of the current time step ($t \mod L$), which is also sufficient as well as compact information for optimal $c$-persistent decision-making, showing better performance than SAC-L in high-dimensional action tasks. Still, it is unable to exploit every transition sample to update every timestep's actor and critic, which leads to learning inefficiency compared to AP-AC. Finally, **AP-AC** significantly outperforms all of the baseline algorithms in all benchmark domains except for Hopper where AP-AC and baselines are on par. The experimental results highlight the effectiveness of our method that directly optimizes a periodic non-stationary policy for the tasks with multiple control frequencies.

## 7 Discussion and Conclusion

In this work, we formalized the notion of *multiple* action persistences in RL, which generalizes the result of [13] that deals with single action persistence. We introduced AP-PI, an efficient tabular planning algorithm for $c$-persistent policy for FA-MDP, and showed a formal analysis on its optimal convergence guarantee while it has only a marginal increase in the time complexity compared to the standard policy iteration. We then presented AP-AC, an off-policy deep reinforcement learning algorithm that scales, which directly exploits the structure of the optimal solution from the formal analysis on AP-PI. We empirically demonstrated that AP-AC significantly outperforms a number of strong baselines, both on continuous and discrete problems with action persistence. Extending the results of this work to multi-agent or hierarchical RL would be an interesting direction for future work.

## Broader Impact

In recent years, reinforcement learning (RL) has shown remarkable successes in various areas, where most of their results are based on the assumption that all decision variables are simultaneously determined at every discrete time step. However, many real-world sequential decision-making problems involve multiple decision variables whose control frequencies are different by the domain requirement. In this situation, standard RL algorithms without considering the control frequency requirement may suffer from severe performance degradation as discussed in Section 3. This paper provides a theoretical and algorithmic foundation of how to address multiple control frequencies in RL, which enables RL to be applied to more complex and diverse real-world problems that involve decision variables with different frequencies. Therefore, this work would be beneficial for those who want to apply RL to various tasks that inherently have multiple control frequencies. As we provide a general-purpose methodology, we believe this work has little to do with a particular system failure or a particular data bias. On the other hand, this work could contribute to accelerating industrial adoption of RL, which has the potential to adversely affect employment due to automation.

## Acknowledgments

This work was supported by the National Research Foundation (NRF) of Korea (NRF-2019R1A2C1087634 and NRF-2019M3F2A1072238), the Ministry of Science and Information communication Technology (MSIT) of Korea (IITP No. 2020-0-00940, IITP No. 2019-0-00075, IITP No. 2017-0-01779 XAI), and POSCO.

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
