[Supplementary Material · supplementary.pdf]



## Appendix A   Proof of Theorem 1

**Lemma 1.** *For any $\pi \in \Pi_L$, $t \in \{0, \dots, L-1\}$, and $k \in \mathbb{N}$, the composition of $k$ one-step $c$-persistent Bellman operators $(\bar{\mathcal{T}}_t^\pi \cdots \bar{\mathcal{T}}_{(t+k-1) \bmod L}^\pi)$ satisfies:*

$$(\bar{\mathcal{T}}_t^\pi \cdots \bar{\mathcal{T}}_{(t+k-1) \bmod L}^\pi Q)(s,a) = \mathbb{E}_{\forall \tau, s_{\tau+1} \sim P(\cdot|s_\tau, \bar{a}_\tau)} \left[ \sum_{\tau=t}^{t+k-1} \gamma^{\tau-t} R(s_\tau, \bar{a}_\tau) + \gamma^k Q(s_{t+k}, \bar{a}_{t+k}) \;\middle|\; s_t = s, \, \bar{a}_t = a \right]$$

$$\text{where } \bar{a}_{\tau+1} = \Gamma_{\tau+1, \bar{a}_\tau}^c(\pi_{(\tau+1) \bmod L}(\bar{a}_\tau, s_{\tau+1})) \text{ for } \tau = t, \dots, t+k-1$$

*Proof.* We give a proof based on induction. For $k = 1$,

$$(\bar{\mathcal{T}}_t^\pi Q)(s,a) = \mathbb{E}_{s_{t+1} \sim P(\cdot|s_t, \bar{a}_t)} \left[ R(s_t, \bar{a}_t) + \gamma Q(s_{t+1}, \bar{a}_{t+1}) \mid s_t = s, \bar{a}_t = a \right]$$

$$\text{where } \bar{a}_{t+1} = \Gamma_{t+1, \bar{a}_t}^c(\pi_{(t+1) \bmod L}(\bar{a}_t, s_{t+1}))$$

holds by the definition of one-step $c$-persistent Bellman operator $\bar{\mathcal{T}}_t^\pi$ (Eq. (5)). Now, assume the induction hypothesis for $k$. Then,

$$\left( \bar{\mathcal{T}}_t^\pi \cdots \bar{\mathcal{T}}_{(t+k-1) \bmod L}^\pi (\bar{\mathcal{T}}_{(t+k) \bmod L}^\pi Q) \right)(s,a)$$

$$= \mathbb{E}_{\forall \tau, s_{\tau+1} \sim P(\cdot|s_\tau, \bar{a}_\tau)} \left[ \sum_{\tau=t}^{t+k-1} \gamma^{\tau-t} R(s_\tau, \bar{a}_\tau) + \gamma^k (\bar{\mathcal{T}}_{(t+k) \bmod L}^\pi Q)(s_{t+k}, \bar{a}_{t+k}) \;\middle|\; s_t = s, \, \bar{a}_t = a \right]$$

$$\text{where } \bar{a}_{\tau+1} = \Gamma_{\tau+1, \bar{a}_\tau}^c(\pi_{(\tau+1) \bmod L}(\bar{a}_\tau, s_{\tau+1})) \text{ for } \tau = t, \dots, t+k-1$$

(by the induction hypothesis)

$$= \mathbb{E}_{\forall \tau, s_{\tau+1} \sim P(\cdot|s_\tau, \bar{a}_\tau)} \left[ \sum_{\tau=t}^{t+k-1} \gamma^{\tau-t} R(s_\tau, \bar{a}_\tau) + \gamma^k \Big( {\color{red} R(s_{t+k}, \bar{a}_{t+k})} \right.$$

$$\left. {\color{red} + \gamma Q\big(s_{t+k+1}, \Gamma_{t+k+1, \bar{a}_{t+k}}^c(\pi_{(t+k+1) \bmod L}(\bar{a}_{t+k}, s_{t+k+1}))\big) \Big)} \;\middle|\; s_t = s, \, \bar{a}_t = a \right]$$

$$\text{where } \bar{a}_{\tau+1} = \Gamma_{\tau+1, \bar{a}_\tau}^c(\pi_{(\tau+1) \bmod L}(\bar{a}_\tau, s_{\tau+1})) \text{ for } \tau = t, \dots, t+k-1$$

$$= \mathbb{E}_{\forall \tau, s_{\tau+1} \sim P(\cdot|s_\tau, \bar{a}_\tau)} \left[ \sum_{\tau=t}^{t+k} \gamma^{\tau-t} R(s_\tau, \bar{a}_\tau) + \gamma^{k+1} Q(s_{t+k+1}, \bar{a}_{t+k+1}) \;\middle|\; s_t = s, \, \bar{a}_t = a \right]$$

$$\text{where } \bar{a}_{\tau+1} = \Gamma_{\tau+1, \bar{a}_\tau}^c(\pi_{(\tau+1) \bmod L}(\bar{a}_\tau, s_{\tau+1})) \text{ for } \tau = t, \dots, t+k$$

thus the given statement holds for $k+1$, which concludes the proof. $\square$

**Theorem 1.** *For all $t \in \{0, \dots, L-1\}$, the $L$-step $c$-persistent Bellman operator $\bar{H}_t^\pi$ is $\gamma^L$-contraction with respect to infinity norm, thus $\bar{H}_t^\pi Q_t^{\bar{\pi}} = Q_t^{\bar{\pi}}$ has the unique fixed point solution. In other words, for any $Q_t^0 : \mathcal{S} \times \mathcal{A} \to \mathbb{R}$, define $Q_t^{n+1} = \bar{H}_t^\pi Q_t^n$. Then, the sequence $Q_t^n$ converges to $t$-th $c$-persistent value function of $\bar{\pi}$ as $n \to \infty$.*

*Proof.* By Lemma 1, for any $t, s, a$, and $Q_1 : \mathcal{S} \times \mathcal{A} \to \mathbb{R}$ and $Q_2 : \mathcal{S} \times \mathcal{A} \to \mathbb{R}$,

$$\left| \bar{H}_t^\pi Q_1(s,a) - \bar{H}_t^\pi Q_2(s,a) \right| = \left| \mathbb{E}_{\forall \tau, s_{\tau+1} \sim P(\cdot|s_\tau, \bar{a}_\tau)} \left[ \gamma^L Q_1(s_{t+L}, \bar{a}_{t+L}) - \gamma^L Q_2(s_{t+L}, \bar{a}_{t+L}) \mid s_t = s, \bar{a}_t = a \right] \right|$$

$$\text{where } \bar{a}_{\tau+1} = \Gamma_{\tau+1, \bar{a}_\tau}^c(\pi_{(\tau+1) \bmod L}(\bar{a}_\tau, s_{\tau+1})) \text{ for } \tau = t, \dots, t+L-1$$

$$\leq \gamma^L \max_{s', a'} \left| Q_1(s', a') - Q_2(s', a') \right|$$

$$\therefore \left\| \bar{H}_t^\pi Q_1 - \bar{H}_t^\pi Q_2 \right\|_\infty \leq \gamma^L \left\| Q_1 - Q_2 \right\|_\infty$$

Therefore, $\bar{H}_t^\pi$ is $\gamma^L$-contraction with respect to infinity norm, and by Banach fixed-point theorem, $\bar{H}_t^\pi Q_t^{\bar{\pi}} = Q_t^{\bar{\pi}}$ has the unique fixed point solution for all $t$. $\square$

A deeper discussion on the Bellman operators with a periodic non-stationary policy can be found in [10, 18] though it analyzes the error in approximate policy/value iterations, rather than considering action persistence.

**Corollary 1.** $Q_t^{\bar{\pi}} = \bar{\mathcal{T}}_t^{\pi} Q_{(t+1) \bmod L}^{\bar{\pi}}$ *holds for all* $t \in \{0, \dots, L-1\}$, *thus c-persistent value functions can be obtained by repeatedly applying 1-step c-persistent backup in a L-cyclic manner.*

*Proof.*

$$
\begin{aligned}
Q_t^{\bar{\pi}} &= \bar{H}_t^{\pi} Q_t^{\bar{\pi}} = \bar{H}_t^{\pi} \bar{H}_t^{\pi} Q_t^{\bar{\pi}} = \bar{H}_t^{\pi} \bar{H}_t^{\pi} \bar{H}_t^{\pi} Q_t^{\bar{\pi}} = \cdots \qquad \text{(by Theorem 1)} \\
&= \underbrace{\bar{\mathcal{T}}_t^{\pi} \bar{\mathcal{T}}_{(t+1) \bmod L}^{\pi} \cdots \bar{\mathcal{T}}_{(t+L-1) \bmod L}^{\pi}}_{\bar{H}_t^{\pi}} \underbrace{\bar{\mathcal{T}}_t^{\pi} \bar{\mathcal{T}}_{(t+1) \bmod L}^{\pi} \cdots \bar{\mathcal{T}}_{(t+L-1) \bmod L}^{\pi}}_{\bar{H}_t^{\pi}} Q_t^{\bar{\pi}} \\
&= \bar{\mathcal{T}}_t^{\pi} \underbrace{\bar{\mathcal{T}}_{(t+1) \bmod L}^{\pi} \cdots \bar{\mathcal{T}}_{(t+L-1) \bmod L}^{\pi} \bar{\mathcal{T}}_t^{\pi}}_{\bar{H}_{(t+1) \bmod L}^{\pi}} \underbrace{\bar{\mathcal{T}}_{(t+1) \bmod L}^{\pi} \cdots \bar{\mathcal{T}}_{(t+L-1) \bmod L}^{\pi} Q_t^{\bar{\pi}}}_{\triangleq Q} \\
&= \bar{\mathcal{T}}_t^{\pi} \bar{H}_{(t+1) \bmod L}^{\pi} Q = \cdots = \bar{\mathcal{T}}_t^{\pi} \lim_{n \to \infty} (\bar{H}_{(t+1) \bmod L}^{\pi})^n Q \\
&= \bar{\mathcal{T}}_t^{\pi} Q_{(t+1) \bmod L}^{\bar{\pi}} \qquad \qquad \qquad \qquad \text{(by Theorem 1)} \qquad \qquad \square
\end{aligned}
$$

## Appendix B   Proof of Theorem 2

**Theorem 2.** *Given a L-periodic, non-stationary, and deterministic policy* $\pi = (\pi_0, \dots, \pi_{L-1}) \in \Pi_L$, *let* $Q_t^{\bar{\pi}}$ *be the c-persistent value of* $\bar{\pi}$ *denoted in Eq.* (7). *If we update the new policy* $\pi^{\text{new}} = (\pi_0^{\text{new}}, \dots, \pi_{L-1}^{\text{new}}) \in \Pi_L$ *by*

$$
\forall t, a, s', \ \pi_t^{\text{new}}(a, s') = \arg\max_{a'} Q_t^{\bar{\pi}}(s', \Gamma_{t,a}^c(a')) \tag{8}
$$

*then* $Q_t^{\bar{\pi}^{\text{new}}}(s, a) \geq Q_t^{\bar{\pi}}(s, a)$ *holds for all* $t, s, a$.

*Proof.* For any $t, s, a$,

$$
\begin{aligned}
&Q_t^{\bar{\pi}}(s, a) \\
&= \mathbb{E}_P \Big[ R(s_t, \bar{a}_t) + \gamma Q_{(t+1) \bmod L}^{\bar{\pi}} \big( s_{t+1}, \Gamma_{t+1, \bar{a}_t}^c(\pi_{(t+1) \bmod L}(\bar{a}_t, s_{t+1})) \big) \mid s_t = s, \bar{a}_t = a \Big] \\
&\leq \mathbb{E}_P \Big[ R(s_t, \bar{a}_t) + \gamma Q_{(t+1) \bmod L}^{\bar{\pi}} \big( s_{t+1}, \Gamma_{t+1, \bar{a}_t}^c(\pi_{(t+1) \bmod L}^{\text{new}}(\bar{a}_t, s_{t+1})) \big) \mid s_t = s, \bar{a}_t = a \Big] \quad \text{(by Eq. (8))} \\
&= \mathbb{E}_P \Big[ R(s_t, \bar{a}_t) + \gamma \Big( R(s_{t+1}, \bar{a}_{t+1}) + \gamma Q_{(t+2) \bmod L}^{\bar{\pi}} \big( s_{t+2}, \Gamma_{t+2, \bar{a}_{t+1}}^c(\pi_{(t+2) \bmod L}(\bar{a}_{t+1}, s_{t+2})) \big) \Big) \mid s_t = s, \bar{a}_t = a \Big] \\
&\qquad \text{where } \bar{a}_{t+1} = \Gamma_{t+1, \bar{a}_t}^c(\pi_{(t+1) \bmod L}^{\text{new}}(\bar{a}_t, s_{t+1})) \\
&= \mathbb{E}_P \Big[ \sum_{\tau=t}^{t+1} \gamma^{\tau-t} R(s_\tau, \bar{a}_\tau) + \gamma^2 Q_{(t+2) \bmod L}^{\bar{\pi}} \big( s_{t+2}, \Gamma_{t+2, \bar{a}_{t+1}}^c(\pi_{(t+2) \bmod L}(\bar{a}_{t+1}, s_{t+2})) \big) \mid s_t = s, \bar{a}_t = a \Big] \\
&\qquad \text{where } \bar{a}_{\tau+1} = \Gamma_{\tau+1, \bar{a}_\tau}^c(\pi_{(\tau+1) \bmod L}^{\text{new}}(\bar{a}_\tau, s_{\tau+1})) \text{ for } \tau = t \\
&\leq \mathbb{E}_P \Big[ \sum_{\tau=t}^{t+1} \gamma^{\tau-t} R(s_\tau, \bar{a}_\tau) + \gamma^2 Q_{(t+2) \bmod L}^{\bar{\pi}} \big( s_{t+2}, \Gamma_{t+2, \bar{a}_{t+1}}^c(\pi_{(t+2) \bmod L}^{\text{new}}(\bar{a}_{t+1}, s_{t+2})) \big) \mid s_t = s, \bar{a}_t = a \Big] \\
&\qquad \text{where } \bar{a}_{\tau+1} = \Gamma_{\tau+1, \bar{a}_\tau}^c(\pi_{(\tau+1) \bmod L}^{\text{new}}(\bar{a}_\tau, s_{\tau+1})) \text{ for } \tau = t \\
&= \mathbb{E}_P \Big[ \sum_{\tau=t}^{t+2} \gamma^{\tau-t} R(s_\tau, \bar{a}_\tau) + \gamma^3 Q_{(t+3) \bmod L}^{\bar{\pi}} \big( s_{t+3}, \Gamma_{t+3, \bar{a}_{t+2}}^c(\pi_{(t+3) \bmod L}(\bar{a}_{t+2}, s_{t+3})) \big) \mid s_t = s, \bar{a}_t = a \Big] \\
&\qquad \text{where } \bar{a}_{\tau+1} = \Gamma_{\tau+1, \bar{a}_\tau}^c(\pi_{(\tau+1) \bmod L}^{\text{new}}(\bar{a}_\tau, s_{\tau+1})) \text{ for } \tau = t, \ t+1 \\
&\ \ \vdots \\
&\leq \mathbb{E}_P \Big[ \sum_{\tau=t}^{\infty} \gamma^{\tau-t} R(s_\tau, \bar{a}_\tau) \mid s_t = s, \bar{a}_t = a \Big] \\
&\qquad \text{where } \bar{a}_{\tau+1} = \Gamma_{\tau+1, \bar{a}_\tau}^c(\pi_{(\tau+1) \bmod L}^{\text{new}}(\bar{a}_\tau, s_{\tau+1})) \text{ for } \tau = t, \ t+1, \ t+2, \dots \\
&= Q_t^{\bar{\pi}^{\text{new}}}(s, a)
\end{aligned}
$$

where each of inequalities holds by Eq. (8), and this concludes the proof. $\qquad \square$

## Appendix C   Proof of Theorem 3

We first define the following one-step $c$-persistent Bellman *optimality* operator:

$$(\bar{\mathcal{T}}_t^* Q)(s,a) \triangleq \mathbb{E}_{s' \sim P(\cdot|s,a)}\left[R(s,a) + \gamma \max_{a'} Q(s', \Gamma_{t+1,a}^c(a'))\right] \tag{11}$$

Note that the one-step $c$-persistent Bellman optimality operators are $L$-periodic with respect to $t$ due to the $L$-periodic nature of the projection operator $\Gamma_{t,a}^c(a')$. Therefore, $\bar{\mathcal{T}}_t^* Q = \bar{\mathcal{T}}_{t+L}^* Q$ always holds for any $t$ and $Q$. Then, similar to Eq. (6), we define an $L$-step $c$-persistent Bellman *optimality* operator $\bar{H}_t^*$ by making the composition of $L$ one-step $c$-persistent Bellman optimality operators:

$$(\bar{H}_0^* Q)(s,a) \triangleq (\bar{\mathcal{T}}_0^* \bar{\mathcal{T}}_1^* \cdots \bar{\mathcal{T}}_{L-2}^* \bar{\mathcal{T}}_{L-1}^* Q)(s,a) \tag{12}$$

$$(\bar{H}_1^* Q)(s,a) \triangleq (\bar{\mathcal{T}}_1^* \bar{\mathcal{T}}_2^* \cdots \bar{\mathcal{T}}_{L-1}^* \bar{\mathcal{T}}_0^* Q)(s,a)$$

$$\vdots$$

$$(\bar{H}_{L-1}^* Q)(s,a) \triangleq (\bar{\mathcal{T}}_{L-1}^* \bar{\mathcal{T}}_0^* \cdots \bar{\mathcal{T}}_{L-3}^* \bar{\mathcal{T}}_{L-2}^* Q)(s,a)$$

Similar to $L$-step $c$-persistent Bellman operators, we can show that $L$-step $c$-persistent Bellman *optimality* operators are contraction mapping.

**Lemma 2.** *For all $t \in \{0, \ldots, L-1\}$, the $L$-step $c$-persistent Bellman optimality operator $\bar{H}_t^*$ is $\gamma^L$-contraction with respect to infinity norm, thus $\bar{H}_t^* Q_t^{\bar{*}} = Q_t^{\bar{*}}$ has the unique fixed point solution. In other words, for any $Q_t^0 : \mathcal{S} \times \mathcal{A} \to \mathbb{R}$, define $Q_t^{n+1} = \bar{H}_t^* Q_t^n$. Then, the sequence $Q_t^n$ converges to $t$-th $c$-persistent optimal value function as $n \to \infty$.*

*Proof.* Without loss of generality, it is sufficient to prove when $t = 0$.

For any $Q_1 : \mathcal{S} \times \mathcal{A} \to \mathbb{R}$, $Q_2 : \mathcal{S} \times \mathcal{A} \to \mathbb{R}$, and $s_0 \in \mathcal{S}$, $a_0 \in \mathcal{A}$,

$$|(\bar{H}_0^* Q_1)(s_0, a_0) - (\bar{H}_0^* Q_2)(s_0, a_0)|$$

$$= |(\bar{\mathcal{T}}_0^* \bar{\mathcal{T}}_1^* \cdots \bar{\mathcal{T}}_{L-1}^* Q_1)(s_0, a_0) - (\bar{\mathcal{T}}_0^* \bar{\mathcal{T}}_1^* \cdots \bar{\mathcal{T}}_{L-1}^* Q_2)(s_0, a_0)|$$

$$= \left|\mathbb{E}_{s_1 \sim P(\cdot|s_0, a_0)}\left[R(s_0, a_0) + \gamma \max_{a_1}(\bar{\mathcal{T}}_1^* \cdots \bar{\mathcal{T}}_{L-1}^* Q_1)(s_1, \Gamma_{1,a_0}^c(a_1))\right]\right.$$

$$\left. - \mathbb{E}_{s_1 \sim P(\cdot|s_0, a_0)}\left[R(s_0, a_0) + \gamma \max_{a_1}(\bar{\mathcal{T}}_1^* \cdots \bar{\mathcal{T}}_{L-1}^* Q_2)(s_1, \Gamma_{1,a_0}^c(a_1))\right]\right|$$

$$= \gamma\left|\mathbb{E}_P\left[\max_{a_1}(\bar{\mathcal{T}}_1^* \cdots \bar{\mathcal{T}}_{L-1}^* Q_1)(s_1, \Gamma_{1,a_0}^c(a_1)) - \max_{a_1}(\bar{\mathcal{T}}_1^* \cdots \bar{\mathcal{T}}_{L-1}^* Q_2)(s_1, \Gamma_{1,a_0}^c(a_1))\right]\right|$$

$$\leq \gamma\left|\mathbb{E}_P\left[(\bar{\mathcal{T}}_1^* \cdots \bar{\mathcal{T}}_{L-1}^* Q_1)(s_1, a_1^*) - (\bar{\mathcal{T}}_1^* \cdots \bar{\mathcal{T}}_{L-1}^* Q_2)(s_1, a_1^*)\right]\right|$$

$$\text{where } a_1^* = \arg\max_a \left[(\bar{\mathcal{T}}_1^* \cdots \bar{\mathcal{T}}_{L-1}^* Q_1)(s_1, \Gamma_{1,a_0}^c(a)) - (\bar{\mathcal{T}}_1^* \cdots \bar{\mathcal{T}}_{L-1}^* Q_2)(s_1, \Gamma_{1,a_0}^c(a))\right]$$

$$\leq \gamma \max_{s,a}\left|(\bar{\mathcal{T}}_1^* \cdots \bar{\mathcal{T}}_{L-1}^* Q_1)(s, a) - (\bar{\mathcal{T}}_1^* \cdots \bar{\mathcal{T}}_{L-1}^* Q_2)(s, a)\right|$$

We can continue to expand the inequality in a similar way,

$$\forall s_0, a_0, \; |(\bar{H}_0^* Q_1)(s_0, a_0) - (\bar{H}_0^* Q_1)(s_0, a_0)| \leq \gamma \max_{s,a}\left|(\bar{\mathcal{T}}_1^* \cdots \bar{\mathcal{T}}_{L-1}^* Q_1)(s, a) - (\bar{\mathcal{T}}_1^* \cdots \bar{\mathcal{T}}_{L-1}^* Q_2)(s, a)\right|$$

$$\leq \gamma^2 \max_{s,a}\left|(\bar{\mathcal{T}}_2^* \cdots \bar{\mathcal{T}}_{L-1}^* Q_1)(s, a) - (\bar{\mathcal{T}}_2^* \cdots \bar{\mathcal{T}}_{L-1}^* Q_2)(s, a)\right|$$

$$\vdots$$

$$\leq \gamma^L \max_{s,a}\left|Q_1(s, a) - Q_2(s, a)\right|$$

$$\therefore \|\bar{H}_0^* Q_1 - \bar{H}_0^* Q_2\|_\infty \leq \gamma^L \|Q_1 - Q_2\|_\infty$$

Therefore, $\bar{H}_t^*$ is $\gamma^L$-contraction with respect to infinity norm, and by Banach fixed-point theorem, $\bar{H}_t^* Q_t^{\bar{*}} = Q_t^{\bar{*}}$ has the unique fixed point solution for all $t$. □

Therefore, the optimal $c$-persistent value functions (i.e. the fixed points of each $\bar{H}_0^*, \ldots \bar{H}_{L-1}^*$) can be represented by $L$ values, $(Q_0^{\bar*}, \ldots, Q_{L-1}^{\bar*})$. Also, the following lemma shows that they have the largest possible value, compared to any $c$-persistent value functions of any history-dependent policy $\pi \in \Pi$.

**Lemma 3.** *For any $t$, let $\bar{H}_{t \bmod L}^* = \bar{\mathcal{T}}_{t \bmod L}^* \ldots \bar{\mathcal{T}}_{(t+L-1) \bmod L}^*$ be $L$-step $c$-persistent Bellman optimality operator and $Q_{t \bmod L}^{\bar*}$ be its fixed point. Then, for any history-dependent policy $\pi \in \Pi$, $Q_{t \bmod L}^{\bar*}(s,a) \geq Q_t^{\bar\pi}(s,a)$ holds for all $t, s, a$.*

*Proof.* For any $\pi \in \Pi$, $t \in \mathbb{N}_0$, $s \in \mathcal{S}$, $a \in \mathcal{A}$, and $Q : \mathcal{S} \times \mathcal{A} \to \mathbb{R}$, the following inequality holds:

$$
\begin{aligned}
(\bar{\mathcal{T}}_t^\pi Q)(s_t, a_t) &\triangleq R(s_t, a_t) + \gamma \mathbb{E}_{\substack{s_{t+1} \sim P(\cdot|s_t,a_t) \\ a_{t+1} \sim \pi(\cdot|h_{t+1})}} \left[ Q(s_{t+1}, \Gamma_{t+1,a_t}^c(a_{t+1})) \right] \\
&\leq R(s_t, a_t) + \gamma \max_{a'} \mathbb{E}_{s_{t+1} \sim P(\cdot|s_t,a_t)} \left[ Q(s_{t+1}, \Gamma_{t+1,a_t}^c(a')) \right] \\
&= (\bar{\mathcal{T}}_{t \bmod L}^* Q)(s_t, a_t)
\end{aligned}
$$

which implies

$$
(\bar{\mathcal{T}}_t^\pi \bar{\mathcal{T}}_{t+1}^\pi \ldots \bar{\mathcal{T}}_{t+L-1}^\pi Q)(s,a) \leq (\bar{\mathcal{T}}_{t \bmod L}^* \bar{\mathcal{T}}_{(t+1) \bmod L}^* \ldots \bar{\mathcal{T}}_{(t+L-1) \bmod L}^* Q)(s,a) = (\bar{H}_{t \bmod L}^* Q)(s,a)
$$

Therefore, $Q_t^{\bar\pi}(s,a) = \lim_{n\to\infty}(\bar{\mathcal{T}}_t^\pi \bar{\mathcal{T}}_{t+1}^\pi \ldots \bar{\mathcal{T}}_{t+Ln-1}^\pi Q)(s,a) \leq \lim_{n\to\infty}((\bar{H}_{t \bmod L}^*)^n Q)(s,a) = Q_{t \bmod L}^{\bar*}(s,a)$ holds for any $t, s, a$ and history-dependent policy $\pi$, which concludes the proof. $\square$

Now, we are ready to provide the proof of Theorem 3.

**Theorem 3.** *Starting from any $\bar\pi^0 \in \Pi_c$ induced by $L$-periodic non-stationary deterministic policy $\pi^0 \in \Pi_L$, the sequence of value functions $Q^{\bar\pi^n}$ and the improved policies $\bar\pi^{n+1}$ induced by $\pi^{n+1}$ converge to the optimal value function and the optimal $c$-persistent policy $\bar\pi^*$, i.e. $Q_t^{\bar*}(s,a) = \lim_{n\to\infty} Q_{t \bmod L}^{\bar\pi^n}(s,a) \geq Q_t^{\bar\pi}(s,a)$ for any $\bar\pi \in \Pi_c$, $t \in \mathbb{N}_0$, $s \in \mathcal{S}$, and $a \in \mathcal{A}$.*

*Proof.* By Lemma 3, it is sufficient to show $\lim_{n\to\infty} Q_t^{\bar\pi^n} = Q_t^{\bar*}$ for all $t \in \{0, \ldots, L-1\}$. By Theorem 2, the performance of $c$-persistent policy induced by $\pi^n$ is monotonically improved during policy iteration, i.e. $Q_t^{\bar\pi^{n+1}}(s,a) \geq Q_t^{\bar\pi^n}(s,a)$ always holds for all $t, s, a, n$. Now, consider when the policy is no longer improved, i.e. $\bar\pi^{n+1} = \bar\pi^n$ and $Q_t^{\bar\pi^{n+1}} = Q_t^{\bar\pi^n}$. In this situation, for all $t, s, a$,

$$
\begin{aligned}
Q_t^{\bar\pi^n}(s,a) &= Q_t^{\bar\pi^{n+1}}(s,a) \\
&= R(s,a) + \gamma \mathbb{E}_{s' \sim P(\cdot|s,a)} \left[ Q_{(t+1) \bmod L}^{\bar\pi^{n+1}}(s', \Gamma_{t+1,a}^c(\bar\pi^{n+1}(a,s'))) \right] \\
&= R(s,a) + \gamma \mathbb{E}_{s' \sim P(\cdot|s,a)} \left[ Q_{(t+1) \bmod L}^{\bar\pi^n}(s', \Gamma_{t+1,a}^c(\bar\pi^{n+1}(a,s'))) \right] \\
&= R(s,a) + \gamma \max_{a'} \mathbb{E}_{s' \sim P(\cdot|s,a)} \left[ Q_{(t+1) \bmod L}^{\bar\pi^n}(s', \Gamma_{t+1,a}^c(a')) \right] \qquad \text{(by Eq. (8))}
\end{aligned}
$$

holds, and this implies that $Q_t^{\bar\pi^n}$ satisfies the $c$-persistent Bellman optimality equation. By Lemma 2, the $c$-persistent Bellman optimality equation has the unique solution, thus $Q_t^{\bar\pi^n} = Q_t^{\bar*}$. This concludes that $\bar\pi^n$ is the optimal $c$-persistent policy. $\square$

**Corollary 2.** *There always exists a $c$-persistent optimal policy $\bar\pi_c^*$, which is induced by a $L$-periodic, non-stationary, and deterministic policy $\pi \in \Pi_L$.*

*Proof.* Every policy $\pi^n$ encountered during action-persistent policy iteration is within $\Pi_L$. Also, by Theorem 3, $\bar\pi^n \in \Pi_c$ induced by $\pi^n \in \Pi_L$ eventually converges to the optimal $c$-persistent policy, which concludes the proof. $\square$

Corollary 2 ensures that the optimal $c$-persistent policy can be always found only through $(\pi_0, \ldots, \pi_{L-1}) \in \Pi_L$, where $\pi_t : \mathcal{A} \times \mathcal{S} \to \mathcal{A}$.

# Appendix D   Pseudo-code of Action-Persistent Policy Iteration (AP-PI)

---

**Algorithm 1** Action-Persistent Policy Iteration (AP-PI)

---

**Input:** $\mathcal{M}$: FA-MDP, $c$: action persistence vector
    Randomly initialize $\pi = (\pi_0, \dots, \pi_{L-1})$ where $\pi_t : \mathcal{A} \times \mathcal{S} \to \mathcal{A}$ for all $t = 0, \dots, L-1$.
    Randomly initialize $Q = (Q_0, \dots, Q_{L-1})$ where $Q_t : \mathcal{S} \times \mathcal{A} \to \mathbb{R}$ for all $t = 0, \dots, L-1$.
    **repeat**
        # Policy Evaluation
        **repeat**
            **for** $t = 0, \dots, L-1$ **do**
                $\forall s, a,\ Q_t(s, a) \leftarrow R(s, a) + \gamma \mathbb{E}_{s' \sim P(\cdot|s,a)} \left[ Q_{(t+1) \bmod L}(s', a') \right]$
                    where $a' = \Gamma^c_{t+1,a}(\pi_{(t+1) \bmod L}(a, s'))$
            **end for**
        **until** $Q$ is converged
        # Policy Improvement
        $\forall t, a, s',\ \pi_t(a, s') \leftarrow \arg\max_{a'} Q_t(s', \Gamma^c_{t,a}(a'))$
    **until** $\pi$ does not change
**Output:** $\bar{\pi}^* = \bar{\pi}$

---

The policy $\bar{\pi}^* = (\bar{\pi}^*_0, \dots, \bar{\pi}^*_{L-1})$ obtained by AP-PI is executed as follows. First, $\bar{a}$ is initialized randomly. Then, at every step $t$, $a_t = \Gamma^c_{t,\bar{a}}(\bar{\pi}^*_{t \bmod L}(\bar{a}, s_t))$ is executed, and the reward and the next state is observed: $r_t, s_{t+1} \sim p(r_t, s_{t+1}|s_t, a_t)$. Finally, $\bar{a}$ is updated by $\bar{a} \leftarrow a_t$, and this procedure continues.

# Appendix E   Pseudo-Code of Action-Persistent Actor-Critic

---

**Algorithm 2** Action-Persistent Actor-Critic (AP-AC)

---

**Input:** $\theta_1, \theta_2, \phi$                                                      ▷ Initialize parameters
    $\bar{\theta}_1 \leftarrow \theta_1$ and $\bar{\theta}_2 \leftarrow \theta_2$               ▷ Initialize target network weights
    $\bar{a} \sim \text{unif}(\mathcal{A})$                            ▷ Initialize $\bar{a}$ randomly
    $\mathcal{D} \leftarrow \emptyset$                 ▷ Initialize a replay buffer to an empty set
    **for** each iteration **do**
        **for** each environment step **do**
            $a_t \sim \pi_{\phi,t}(\cdot|\bar{a}, s_t)$      ▷ Sample a non-persistent action from the policy
            $\bar{a} \leftarrow \Gamma_{t,\bar{a}}(a_t)$            ▷ Project the sampled action using Eq. (4)
            $r_t, s_{t+1} \sim p(r_t, s_{t+1}|s_t, \bar{a})$   ▷ Sample reward and transition from the environment
            $\mathcal{D} \leftarrow \mathcal{D} \cup \{(s_t, \bar{a}, r_t, s_{t+1})\}$   ▷ Store the sampled reward and transition into replay buffer
        **end for**
        **for** each gradient step **do**
            $\theta_i \leftarrow \theta_i - \lambda_Q \hat{\nabla}_{\theta_i} J_Q(\theta_i)$ for $i \in \{1, 2\}$   ▷ Update critic weights by minimizing Eq. (9)
            $\phi \leftarrow \phi + \lambda_\pi \hat{\nabla}_\phi J_\pi(\phi)$         ▷ Update policy weights by maximizing Eq. (10)
            $\bar{\theta}_i \leftarrow \tau\theta_i + (1-\tau)\bar{\theta}_i$ for $i \in \{1, 2\}$     ▷ Update target network weights
        **end for**
    **end for**
**Output:** $\theta_1, \theta_2, \phi$                                   ▷ Optimized parameters

---

# Appendix F    Supplementary Experiments

## F.1    Results on SAC

Figure 6: Results on SAC.

In Figure 5, the baseline SAC agent is trained on the standard **non-persistent** environments while being evaluated on **c-persistent** environments where the action-persistence is enforced. As shown in Figure 6[1], the performance of SAC consistently improves in the non-persistent environment that the agent is trained on, but its naïve projection into a $c$-persistent policy completely fails since the agent *never* considers the action-persistence during training.

## F.2    Effects of varying $c$

Figure 7: Effects of varying action-persistence $c$.

The goal of this work is to provide an efficient solution method for the *given* action-persistence $c$, not finding a proper $c$ to speed up learning. Still, we conducted additional experiments to present the effects on the resulting policy of varying $c$. As can be seen from Figure 7, larger action persistence yields more degradation of asymptotic performance due to a limited degree of freedom of control. AP-AC consistently works well for various $c$'s.

# Appendix G   Experimental Setup

## G.1   Computing Infrastructure

All experiments were conducted on Google Cloud Platform. Specifically, we used the compute-optimized machines (c2-standard-4) that provide 4 vCPUS and 16GB memory.

## G.2   Hyperparameters

Table 1: AP-AC Hyperaparameters

| Parameter | Value |
|---|---|
| optimizer | Adam [6] |
| learning rate | $3 \cdot 10^{-4}$ |
| discount factor $\gamma$ | 0.99 |
| replay buffer size | $10^6$ |
| number of hidden layers (all networks) | 2 |
| number of hidden units per layer | 100 |
| number of samples per minibatch | 100 |
| nonlinearity | ReLU |
| target smoothing coefficient $\tau$ | 0.005 |
| target update interval | 1 |
| gradient steps | 1 |
| (discrete only) temperature of relaxed categorical | 0.1 |

The hyperparameters we used in the experiments are listed in Table 1. Also, for Mujoco continuous control tasks, we used automatic entropy adjustment with the entropy target $-\dim(\mathcal{A})$, and for the discrete action task (i.e. traffic light control), we used the fixed entropy coefficient $\alpha = 0.01$. We simply tried the listed hyperparameters and not tuned them further.

## Footnotes

[1]A performance gap exists compared to those reported in the original SAC paper, due to usage of different hyperparameters such as the number of hidden units per layer, i.e. 100 (ours) / 256 (original SAC paper).