[Reviews · NeurIPS 2020]

Review 1

Summary and Contributions: [Reinforcement Learning for Control with Multiple Frequencies] In this paper, the authors propose a variant of SAC where actions are with different frequency. The authors provide mathematics proof which will help to deepen the understanding of the algorithm. The experiment section indicates that the proposed algorithm can obtain better performance.

Strengths: 1. The algorithm is interesting and studies a real-life problem that could be beneficial to society. 2. The theory is provided for the proposed algorithm.

Weaknesses: The experiment section is weak. The baseline does not seem strong enough. In Figure 5, the performance at 1 million steps for cheetah is only around 3000, and SAC, one baseline that should reach around 10000 reward at 1 million steps, is not learning anything. It seems that the baselines are broken. It also occurs quite obviously for all other environments (Hopper, Walker, Ant).

Correctness: yes

Clarity: yes

Relation to Prior Work: Yes, the related work is adequate.

Reproducibility: Yes

Additional Feedback: I believe I cannot fully understand the paper. So I intend to make the vote for rejection to this project based on the quality of the experiment section. The current paper raises some questions on the quality of the experiment section, which makes the claim of the paper not supported. -----------------------------------AFTER REBUTTAL---------------------------------------- After reading the rebuttal, where the authors have addressed my misunderstanding of the problem setting, as well as discussing with other reviewers, I would like to increase my score to 6 marginal acceptance.


Review 2

Summary and Contributions: This work introduces an algorithm for reinforcement learning in settings with factored action spaces in which each element of the action space may have a different control frequency. To motivate the necessity of such an algorithm, it provides an argument that in this setting, a naive approach with a stationary Markovian policy on the states (which does not observe the timestep) can be suboptimal. Further, it argues that simply augmenting the state or action spaces and applying standard RL methods results in costs which are exponential in L, the least common multiple of the set of action persistences. In constructing the method this paper introduces c-persistent Bellman operators, a way of updating a Q-function in an environment with multiple action persistences, and proves its convergence. This leads to a method which uses L Q-functions, one for each step in the periodic structure of action persistences. Using these Q-functions, the paper introduces a policy improvement step and proves that it does, in fact, improve improve performance. Finally it shows that a policy iteration algorithm based on these components converges to the optimal policy. The paper proposes a practical implementation of these ideas comprising a neural network architecture and an actor-critic algorithm and validates its performance experimentally. **Post rebuttal** Thanks to the authors for the response! The clarification and experiments for the alternate baseline are very helpful. I think this paper does a meticulous job on a problem that's worth solving and I'd like to see it at NeurIPS this year.

Strengths: While the problem studied here is relatively niche within the reinforcement learning community, in practical systems it is common to have multiple control frequencies. This paper tackles the problem in a rigorous way and I could see its methods being used in the future. The algorithm that it proposes is not surprising but the work starts from a real problem, builds up the fundamentals, then solves it in a convincing way. It is refreshing to see a simple thing done very thoroughly. I do not know the work on this particular subproblem well enough to definitively state the novelty of this work, but as someone who has worked on related problems in the past it is new to me. The trick of using every timestep to train every head of the model (line 195) is particularly clever, and should be adopted in the hierarchical and temporally-abstract actions literature more generally. (I wish I had thought of it myself while working on abstract actions last year — it would have directly applied and improved the sample efficiency of a method I developed.) The claims and their proofs appear sound and the proposed algorithm is practical.

Weaknesses: The main weakness of this work as it stands is its treatment of the alternatives. The simplest solution to this problem, which I believe would have all the same properties, would be to condition the value function and the policy on the last action which was _actually taken_ and t mod L. This could be viewed as a modification to the environment rather than to the learning algorithm: the observations from the environment would simply include the last (realized, in the sense of 𝚪) action taken and the current point in the action period. This modification would make the environment satisfy the Markov property once again and permit the straightforward application of unmodified RL methods. Instead of having exponential complexity in L, as was suggested for the alternatives in this work, it would have linear complexity just like the proposed method. This environment modification method would not have all the advantages of the one proposed in this paper, in particular being able to update every timestep with every transition. However, the use of such a method seems less dire than this work implies. Is my understanding correct? Or am I missing something important?

Correctness: Yes. My only wish is for experiments on an environment not proposed by the authors, whether a standard benchmark or a task from the real world. However I don't expect that such a benchmark exists and it the baselines used here seem largely appropriate (though see the Weaknesses section above).

Clarity: The writing is clear and well structured.

Relation to Prior Work: The literature on semi-Markov decision processes seems relevant to this work. Also relevant would be work on hierarchical or temporally abstract action spaces such as "Near-optimal representation learning for hierarchical reinforcement learning" or "Dynamics-Aware Embeddings"; both of these works include off-policy algorithms for temporally abstract actions.

Reproducibility: Yes

Additional Feedback: Thanks for the thorough work!


Review 3

Summary and Contributions: This paper introduces multiple action persistences in RL, where each factored action is repeated for a certain number of steps with its own frequency, and proposes a new policy iteration that guarantees contraction and convergence to the optimal policy. The proposed algorithm was applied to soft actor-critic (SAC), and the results on MuJoCo and traffic control domain show that the proposed method (AP-AC) outperforms the naive stationary policy baseline (SAC) that is unaware of action persistence and other baselines that are aware of action persistence.

Strengths: - The notion of multiple action persistences in FA-MDPs is a nice generalization of the previous work [11]. Although this is not a popular topic, it is worth discussing in that real-world applications may require such constraints (multiple action persistence). So, I think this work is relevant to the RL community. - The proposed policy iteration algorithm is novel and is supported by the theoretical results. Also, its application to advanced methods such as SAC is implemented well. - The empirical result looks good, and the baselines are well-designed.

Weaknesses: - Although the proposed policy iteration is novel, it feels a bit like a straightforward extension of the previous work on Persistent Fitted Q-iteration [11]. It would be good to motivate why it is a non-trivial extension from the previous work and what is the new challenges. - The experimental results could be more comprehensive. For instance, it would be interesting to see how action persistence affects the performance by varying it. It would be also interesting to show some qualitative examples (e.g., traffic control) highlighting the limitation of the naive approaches in contrast to the proposed method.

Correctness: - The theoretical results look intuitively correct, but I have not fully checked the proofs in the appendix.

Clarity: - The paper is well-written. - The motivating example (Figure 1) was helpful for understanding. - Figure 2 is dense with notations, which is not easy to understand.

Relation to Prior Work: - The paper clearly describes the difference between the proposed approach and the relevant previous work [11]. However, it would be good to have a related work section and describe a broader line of work such as semi-MDPs and studies on action-repeats.

Reproducibility: Yes

Additional Feedback: Please see the weakness section for the suggestions.


Review 4

Summary and Contributions: This paper introduces a practical algorithms for solving a particular class of problems where the MDP has factorized action spaces each factor of which are controlled via a fixed frequency. The algorithm derived is theoretically sound and to the best of my knowledge is novel. The authors conducted experiments on simulated environments that show their method performing well against a few baselines.

Strengths: This paper introduces a cute idea on solving MDP with factored persistent actions. On this problem, the proposed approach is very simple to work with and seems to be quite competitive. The paper is also theoretically grounded.

Weaknesses: - It is not very clear whether the proposed approach has many potential areas of application. The setting of the problem seems somewhat restrictive. That being said, I am not an expert in this area. - In all experiments conducted, the least common multiple is small. It is unclear whether the proposed method could generalize to much higher values of least common multiples. In addition, it is not clear whether the proposed algorithm would remain competitive against SAC-L in settings where L is large since in this case the large number of actor and critic heads could hurt performance. If the control frequencies are say (3, 5, 7), the least common multiple would be very high in value. It would be very interesting to see how well the proposed approach as well as the baselines handle this scenario. - "SAC" is a very weak baseline in this setting as it is trained essentially on a different environment. I think this baseline should not be introduced in the main text as it could mislead the readers. It is, however, good to be included in the appendix. - The paper lacks a bit of background in that it does not talk enough about related work.

Correctness: As far as I know the paper makes sound theoretical claims. The experiments seem correct as well.

Clarity: The paper is clearly written.

Relation to Prior Work: This work spent little time talking about previous research. I would be great if the authors could spend more time talking about related work. For example, one could spend more time talking about L-Markovian policy in section 3. More discussion could be had when it comes to [11] which the authors spend little time talking about.

Reproducibility: Yes

Additional Feedback:

[Author Response · NeurIPS 2020]

We thank all the reviewers for their valuable comments. We will reflect the comments in the final version of the paper.

**[R2: results on SAC]** Please let us correct the reviewer's misunderstanding of our experimental results on SAC. In
Figure 5 of the paper, the baseline SAC agent is trained on the standard **non-persistent** environments while being
evaluated on **c-persistent** environments where the action-persistence is enforced. As shown in the following figure[1],
the performance of SAC consistently improves in the non-persistent environment that the agent is trained on, but its
naïve projection into a $c$-persistent policy completely fails since the agent *never* considers the action-persistence during
training. We would like to emphasize this is natural and **not** a broken result.

**[R3: alternative baseline]** Your understanding is correct. The environment modification method, which includes the
'last action' and '$t \bmod L$' in the augmented state, will also enjoy a linear complexity with respect to $L$. However, this
alternative baseline still has some drawbacks compared to our proposed method. First, it is unable to exploit every
transition sample to update every timestep's actor and critic, while AP-AC is capable of doing it in Eq. (9-10). Second,
there exists a redundancy in the representation of $Q$-function, i.e. $Q(t, \bar{a}_{\text{last}}, s, a)$ is not succinct compared to ours of
$Q(t, s, a)$. This incurs a factor of $|\mathcal{A}|^2$ increase in time complexity of policy evaluation in tabular FA-MDPs. Still, we
conducted additional experiments that compare the proposed baseline (i.e. training vanilla SAC agent in the augmented
environment that includes $\bar{a}_{\text{last}}$ and $t \bmod L$ in the observation) to AP-AC. As the following figure demonstrates,
AP-AC still performs better than or on par with the alternative baseline.

**[R3: experiments on standard benchmarks]** To the best of our knowledge, there is no standard benchmark to evaluate
RL algorithms with multiple action persistence. Adopting our method to real-world tasks remains as future work.

**[R3,R4: related works]** Thanks for your suggestion. If every action variable's control frequency is same, $c$-persistent
action can be understood as a particular instance of an option in semi-MDP framework, which always lasts $c$ time steps.
We will add more discussions on related works such as semi-MDPs, temporally abstract actions, and action-repeats in
the final version of the paper.

**[R4: comparison to Persistent FQI (PFQI)]** Our algorithm, AP-AC, is a non-trivial extension of PFQI. First, PFQI
is only applicable to *single* action-persistence and finite action space, while AP-AC can deal with arbitrarily *multiple*
action-persistence and both finite and continuous action spaces. Second, PFQI maintains only one $Q$-function and
performs Bellman optimality backup followed by action-persistence backup $k$ times. Each optimality (or persistence)
backup operation requires to solve a regression problem until convergence. As a consequence, it can only work in
the batch RL (a.k.a. offline RL) setting. Also, as long as PFQI maintains single $Q$-function, its extension to online
RL is not straightforward. In contrast, AP-AC uses $L$ actors and $L$ critics and simultaneously updates all of them via
exploiting their recursive relationship, which enables online learning of the agent as well.

**[R4: more comprehensive experiments]** The goal of this work is to provide an efficient solution method for the *given*
action-persistence $c$, not finding a proper $c$ to speed up learning. Still, we conducted additional experiments to present
the effects on the resulting policy of varying $c$. As the following figure shows, larger action persistence yields more
degradation of asymptotic performance due to a limited degree of freedom of control. AP-AC consistently works well
for various $c$'s. We will also include qualitative examples (e.g. videos) in the supplementary material of the final paper.

We will also place the Broader Impact section into the main text, which is currently in the supplementary material.

## Footnotes

[1]A performance gap exists compared to those reported in the original SAC paper, due to usage of different hyperparameters such as the number of hidden units per layer, i.e. 100 (ours) / 256 (original SAC paper).


[Meta-Review · NeurIPS 2020]

The paper proposes an off-policy policy iteration scheme for factored action spaces where different actions (action dimensions) are persistent with different frequencies. The reviewers agree that the proposed approach is sound, novel, and well motivated. The paper is well written. There is some disagreement how broad the range of applications is to which the proposed method can be applied and what this means for the impact of the paper (R5); some concerns regarding the scalability (R5) of the approach; and some desire for environments not designed by the authors (R2). The AC believes that, although the application domain may be somewhat niche, and the proposed method the result of a somewhat straightforward reasoning about basic properties of MDPs (I don't mean this in a bad way; such basic ideas are often overlooked), on balance the paper will be useful and of interest to the community. The authors are encouraged to take into account the feedback from the reviewers and expand the discussion of related work; the lack of a discussion of semi-MDP seems like an important oversight. (Intra-option learning may be of particular relevance to the off-policy learning scheme proposed by the authors.) Furthermore, the authors are encouraged to address the scalability concerns expressed by R5.